# Fighting for What? Couples' Communication, Parenting and Social Activism: The Case Study of a "Christian-Muslim" Families' Association in Brussels (Belgium)

**Francesco Cerchiaro**

Center for Sociological Research, KU Leuven, 3000 Leuven, Belgium; francesco.cerchiaro@kuleuven.be

**Abstract:** Mixed families have historically been considered to be a direct consequence of a process of social and cultural integration of migrants within the host society, although this link has recently been problematized by scholars. By focusing on the case study of an association of "Christian-Muslim" families in Belgium, this article offers a better understanding of the social consequences of mixedness. The article seeks to shed light on the private and public life of the couples who are members of this association by answering the following research questions: Why do couples turn to this association? At what stages of their lives? What is the social role that the association aims to play in society? Using partners' life stories and ethnographic observation gathered during the association's meetings, the findings demonstrate how this association plays an important role at different levels and at different stages of a family's life. The analysis will highlight that: (1) there is a specific aim to help new couples to face administrative, religious and cultural "obstacles" they encounter during the first period of their relationships, and (2) special meetings to discuss the challenge of parenting are at the core of the association's activities. The "problem" of transmission requires of the couple further negotiations to find a way to balance their respective cultural backgrounds. These negotiations have to take into account the power misbalance within the Belgian hegemonic context. (3) The social activism of this association is an important aspect of its aims and scope. Some of the couples are active in countering a dominant stereotypical representation of mixed couples. They organize meetings and events to sensitize public opinion on interreligious dialogue, migration issues and the fight against racism. In this way, the association proposes itself as a new peculiar agent of social change in the public sphere.

**Keywords:** Christian-Muslim families; mixed families associations; Muslim in Europe; parenting; racism

## 1. Introduction

Mixed families are a growing global phenomena which exemplify the increase in migration flows and provide us with an opportunity to analyze the everyday processes related to Europe's pluralism. The concept of "conjugal mixedness"[1] thus represents a limitation to the family sphere of a much broader notion of the word "mixedness", which involves the whole of society (Varro 2003; Arweck and Nesbitt 2010; Cerchiaro et al. 2015). The sociology of migration has been historically interested in mixed marriages as the ultimate level of integration of the migrants in the host society (Gordon 1964; Kalmijn 1998; Song 2009) while sociology of religion has been interested in interreligious

---

[1]    I refer to the definition given by Collet (2012, p. 71), who pointed out that "conjugal mixedness is not only a question of different cultures but one of conformity or deviance with regard to social norms".

marriages because it has assumed they are the result of a growing secularization (Voas 2003, 2009; Sherkat 2004). Other studies have demonstrated that these marriages lead to higher divorce rates, suggesting that these are due to communication difficulties and disagreements caused by cultural differences between partners, as well as to hostility from families of origin, religious institutions and society at large (Heaton 1984; Davidson 1998; Sherkat 2004; Parisi 2016). These studies are often based on the poorly defined idea that the term "mixed families" represents some kind of monolithic entity based on the different backgrounds (national or religious) of the spouses. The link between mixed families and a greater level of social, cultural and economic integration has, indeed, recently been questioned and problematized.[2] In particular, we need a better understanding of the social consequences of mixedness. What does it imply for the family members and for the social context in which they live? What kind of integration and secularization do they denote? Organizations of mixed families are, in this sense, arising in different social contexts and trying to become autonomous and visible social actors in the public sphere. Even if this is a signal of the growing social relevance of these families, the roles and impact of these organizations have not been investigated yet. This article seeks to contribute to filling this gap by focusing on the case study of a "Christian-Muslim" families' association active in the Brussels capital region (Belgium). In particular the article seeks to answer the following research questions: Why do couples turn to this association? At what stages of their lives? What is the social role that the association aims to play in the society?

Christian-Muslim couples represent an emblematic case-study within the phenomenon of mixed[3] marriages because, in the social space of the family, they incorporate ethnic and religious differences represented as "strong" and "conflicting" in the public hegemonic discourse (Ata 2003; Bangstad 2004; Allievi 2006; Al-Yousuf 2006; Roer-Strier and Ben Ezra 2006). The position of Christian and Islamic religious authorities, openly suspicious of this type of relationship, and the anti-Muslim discourses of the right-wing political parties that are growing in Europe, are important social factors which create a context of external pressure and hostility, influencing the lives and choices of these families. That is why some of these couples are establishing, or actively participating in mixed family associations that aim to offer both support to the couples themselves and to be present in the social debate on migration and religious pluralism.

In the next paragraph we will briefly frame our contribution within the debate on mixed marriages. In particular our analysis aims to contribute to the wider debate on integration and secularization and, especially, to the debate on the concepts of cultural and religious "dilution" and "loss". The article will then introduce a "methodology and participants" section to help the reader in framing our findings, followed by a section on the aims and scope of this association and its social activities. In the end, through partners' life stories, the article will deepen our understanding of how couples deal with religious pluralism in their daily family life. In light of the research results, the conclusions discuss the relevance of our findings for a wider public interested in the debate on mixed couples, integration and secularization.

---

[2]  I refer, in particular, to two special journal issues on mixed marriages: Vol. 662 (1) of The ANNALS of the American Academy of Political and Social Science (2015) and Vol. 14 (4) of Ethnicities (2016). The single contributions are mentioned in the article and listed in the bibliography.

[3]  We cannot debate extensively here the controversies and ambiguities related to the language used to categorize and analyze the phenomenon, but we will briefly explain the decision to employ the term "mixed". On the one hand, as Song and Gutierrez (2015a, p. 2) pointed out, "the terms 'mixed' and 'multiracial' [ . . . ] seem to imply the existence of pure 'races' that can combine into a race mixture". On the other hand, the risk of using a more precise term is to attribute an a priori difference to the couple. In accordance with other scholars (Varro 2003; Edwards et al. 2009; Collet 2012), we thus use the term "mixed" to encompass the multiple differences (regarding ethnicity and religion) which characterize the participants of our study. What "mixed" is about becomes exactly the core question on which each study of intermarriage should finally reflect (Cerchiaro 2016a).

## 2. Mixed Marriages, Integration and Secularization

In this paragraph we will offer an overview of the wider debate on mixed marriages, trying to highlight the major quantitative and qualitative studies in order to make them dialogue together. For the former, these couples mainly represent an indicator of integration or assimilation of the foreign partner into the mainstream society, a marker of the distance or proximity between different social groups. For the latter, mixed marriages are above all a "social laboratory" in which to observe the intercultural practices of the family members, the challenges of "mixed" parenting and their children's multiple identities. In western societies, and in particular in Europe, the most common type of mixed marriage is that which takes place between immigrants and natives (even though there are indications that marriages between immigrants of different nationalities are steadily increasing). For this reason, mixed marriages have traditionally been considered to be the most reliable marker of the integration of immigrants into the host society (Merton 1941; Davis 1941; Gordon 1964; Kalmijn 1998). Despite this, the link between mixed marriages and integration has been contested and problematized by various scholars who have argued for the complexity and multidirectional nature of the processes of integration (Safi 2008; Collet 2012, 2015; Song 2009, 2010; Rodriguez-Garcia 2015; Cerchiaro 2016b). Mixed marriages between partners socialized to different faiths have become an interesting perspective to study the connection between migration, religion and families, i.e., how these families transform the relationship between individuals and religious institutions (Voas 2003, 2009; Sherkat 2004). Generally, quantitative studies attribute to partners in interreligious marriages a lower rate of religious participation (Iannaccone 1990), seen, on the one hand, as a source of risk for the respective religious institutions and, on the other, as a problem that involves the stability of marriage itself. In this regard, some studies have analyzed the relationship between the higher divorce rate, the shorter duration of unions and the causes of marital conflicts to explain the greater vulnerability of these couples (Heaton 1984; Davidson 1998; Sherkat 2004; Parisi 2016). Often categorized as "Christian-Muslim" or "Islamic-Christian", the couples who are the object of this research represent, thus, an emblematic case study of "mixedness", since they incorporate two layers of differences: religious, as the two partners are socialized into Islam and Catholicism respectively, and ethnic, as a white partner is married to a non-white partner. Despite this interplay between two levels of difference, in the public, and often in the academic debate, religious difference is assumed to be the primary conflictual dimension between partners (Ata 2003; Bangstad 2004; Allievi 2006; Al-Yousuf 2006; Roer-Strier and Ben Ezra 2006; Saraceno 2007; Alba and Foner 2015). The majority of research on religiously and ethnically mixed marriages is focused on couples' interactions and parenting styles as the main analytical dimension in exploring how partners deal with their heritages (Odasso 2016; Cerchiaro 2017; Varro 2003; Collet 2012; Arweck and Nesbitt 2010; McCarthy 2007; Murad 2005; Parisi 2016). Existing literature indicates that mixed marriages tend to weaken the transmission of religion to the next generation (Voas 2003), and cause the loosening of ethnic identities (Bratter 2007). This article will investigate the supposed "loosening" (Voas 2003) or "dilution" (Song and Gutierrez 2015b) of the parents' religious backgrounds. In order to achieve these goals, partners' life stories represent an important source of "thick description" (Geertz 1973), that will give us significant data to contribute to the debate reported above.

## 3. Methodology and Participants

By choosing a long-established Christian-Muslim association based in Belgium, the article aims to analyze how partners deal with religious pluralism both in the family context and in the wider social milieu. The interviews analyzed draw on data collected between 2018 and 2019 during a Marie Curie post-doctoral individual fellowship with fieldwork in Belgium.[4] The research project is broadly focused on conjugal mixedness among couples where one partner is an immigrant from a majority-Muslim

---

[4] The ongoing project acronym is "ReMix", "Christian-Muslim families dealing with religious pluralism in everyday family life. Religious reconstruction in religiously mixed marriages". Marie Skłodowska-Curie program, Horizon 2020.

country and the other has Belgian heritage. It took into account the heterogeneity of this kind of couple not only in terms of religious self-identification and practices, but also in terms of gender, class and educational differences. The data collected addresses various topics related to religion and ethnicity, but also to parenting strategies and children's self-identification. This article thus focuses only on the couples who are members of this association. It considers both findings from the interviews conducted with partners and ethnographic data gathered during the couples' meetings. The term "partners' life stories" refers to 26 biographical interviews with the partners of 13 nuclear families living in the Brussels region who have taken part in some way in the association's activities/meetings. Interviews were conducted in their homes by using the *récits de vie* approach proposed by Bertaux (1998). Such an approach consists of conducting biographical in-depth interviews in which the aim is, through the use of open questions, to shed light on "the interviewees' world" within their universe of meaning (ibidem). In order to make the respondents feel freer to express themselves, all interviews were conducted individually, in the absence of other family members.[5] In view of the particularities of the Muslim presence in Belgium, characterized as it is by different ethnic groups, the choice of the sample was not limited to a specific ethnic-national group.

The other *corpus* of data relates to the association's meetings. This association is self-defined as an association of couples "islamo-chrétien", created within a Center for Christian-Muslim Relations active in the Brussels region. It was founded in the early 1980s after the Second Vatican Council in light of the greater attention subsequently paid to relations between Christian and non-Christian religions.

It is not our aim here to compare the couples involved in this association with the other couples interviewed but, rather, to shed light on this association through the couples involved in it. These couples demonstrate, indeed, a common search for living religions without focusing on theological contrapositions, differences and obstacles. Even if this is not the place to probe theological questions, it is important to recall the Koranic norm, which affects the orthopraxis, prohibiting a Muslim woman from marrying a non-Muslim man. The Koran (Koran, 5,5) allows Muslim men to marry only women from "the people of the Book" (*kitabiyya*), i.e., Jewish or Christian women. This is because, according to the Koran, only the father's religion can be passed down. Of the 13 couples involved in this association that we interviewed, 12 were constituted by Muslim men and Catholic women and just one by a Muslim woman and a Catholic man. This imbalanced composition, in terms of gender, represents a limitation of the article which does not aim to extensively compare couples where the Muslim partner is the man rather than the woman. The issue of Muslim women married outside their religious group represents even more a challenge which would deserve a dedicated analysis. Muslim women face, indeed, a much harder opposition by their families and Imams, and their marriage cannot be registered in Muslim countries unless the man officially presents a certificate of conversion to Islam.[6] Religiously speaking, I did not find any converted partners or partners who defined themselves as non-religious persons. The very fact of taking part in the association's life was due to the attempt to find a balance between the different religions without losing or changing their own belief. This focus on balance between the partners' faiths seems to emerge as a recurrent statement in the association's meetings, and to be one of the basic objectives of every activity.

Due to the necessity to preserve anonymity, the details and name of the association have been removed from the article, whilst pseudonyms have been used for the participants' names. The difficulty of gaining access to this field and the necessity to gain trust and to assure privacy to the associations and the participants are important elements to take into consideration. They testify, on the one hand, to the context of intimacy and family life that the couples tend to protect from external intrusions, and,

---

[5]　The average duration of each interview was between 2 and 4 hours. All interviews were originally fully transcribed in French.

[6]　An exception is represented by Tunisia where the government in August 2017 abolished the ban for marriages between Muslim women and non-Muslim men.

on the other hand, to the context of stigmatization and external pressure with which these couples have to live.

## 4. Findings

The association's meetings and the partners' interviews offer us meaningful data[7] through which to analyze the organization's activities. Its actions are, indeed, not limited to creating connections among the members but also extend to proposing itself as a social actor that can contribute to the public debate in the wider social context. This findings section seeks analytically to classify the functions that this association carries out. In the first paragraph, "support for couples in the early stages of their relationship", the analysis focuses on the initial support that the partners may need in the first period of their relationship. The second paragraph focuses on the discussions around children's religious education, as one of the main issues with which partners have to deal. The last paragraph analyses the "social activism" and the public actions that the association proposes in order to deepen knowledge of the common ground between Islam and Christianity and to counter the stereotypical dominant representation of Islam.

### 4.1. Support for Couples in the Early Stages of Their Relationship

Both in couples' life stories and in the association's meetings there emerges the importance that the association has in the early stages of a couple's experience. Especially for new couples, the association represents a space for confronting one's initial experience as a couple without feeling judged. It becomes clear that there are some critical aspects common to all couples that are discussed: how to cope with parental interference, administrative issues (such as the registration of the marriage in the country of origin of the migrant partner), the question of the marriage ceremony itself (starting from the choice of civil or religious) and, more generally, the management of religious and cultural difference during their life course. A meaningful example is given in these excerpts from Leen and Hamid, a new couple who was participating for the first time in the couples' meetings.

> We're engaged but not yet married. Actually, this is one of the reasons why we are here. We would like to get married but we do not know many things. How to deal with religious marriage, we are still evaluating. To do civil marriage? Religious marriage? But how? And what about the registration of our marriage in Morocco? It seems there are a lot of complicated issues and we don't know who to ask for an answer. In addition, we would like to discuss with you some things about the culture and the religion of the other. We are also having difficulties with our families. My family, for instance, it's really . . . I would say . . . skeptical about my relationship with Hamid . . . and we would like to talk about it?
>
> Leen, 34, Belgian—from couples' meeting.

> Above all, for the family issue we are a bit under stress. We would like to talk with you also to understand how other couples dealt with similar problems. Then we are discussing about how to do for religions. We are both not very practicing, but here we hold our faith. I feel Muslim and she is Christian. There are points that unite us but also that divide us, even cultural things here . . . on which we have difficulties. And then yes, we ask ourselves for the future, if we had children . . .
>
> Hamid, 35, Morocco—from couples' meeting.

Leen and Hamid, in presenting themselves to the other couples, highlight most of the initial recurrent problems that we then found more deeply in partners' life stories. There is the family

---

[7]  In order to become a member of the association it is usually required to be part of a "Christian-Muslim" couple. I was able to take part in some of their meetings after a long negotiation during which I managed to create a relationship of mutual trust.

opposition (or "skepticism") over the relationship, above all from the Belgian family of the woman. In this sense the association welcomes "young couples"[8] to discuss their initial difficulties, listening to or asking questions of older couples who have already experienced similar situations during their life course. As reported by one of the older leaders of the association when I asked him why a couple should be interested in joining them:

> Here we welcome also couples who just want to ask for information, tips [...] We know the territory, the paper you need if you do not have the resident permit, but also, if you want to marry religiously, we can introduce them to a priest whom you can ask for the dispensation, because the theory is one thing, who gives you permission to marry is another ... or the Imam that is prepared to do the marriage, most of them refuse to celebrate these marriages unless a partner has converted to Islam. [...] And then ... we are all members of a couple. We all know what kind of obstacles you can experience, above all at the beginning of your relationship, with relatives and families, but also later on, over time, with children. Here there were also couples who came just for some months, in their initial period. Then they quit because they knew what they wanted to know ... or because they broke up. But not everyone is necessarily permanently engaged in the associations as I am.

> Abdoullah, Moroccan, 48.

In the first period of their relationship partners can find in this association also a place to gain information on the territory, tips, practical advice on dealing with administrative procedures or names of religious leaders more open to celebrating a mixed marriage. The role of these older couples seems, therefore, to be both that of intermediary between the couple and institutions (both religious and civic) and of a place where they can feel free to express doubts and difficulties with people who are assumed not to be judgmental or biased.

### 4.2. Parenting Advisory

As has been previously analyzed in another article (Cerchiaro et al. 2015), it is especially within the sphere of children's religious education that partners report facing the challenge of religious difference within the family life. This is a choice around parenting which cannot be avoided because even delegating it to the other partner becomes a choice in itself. Parenting thus reveals the strategies adopted by partners to deal with religious pluralism in everyday family life. It is relevant to analyze how partners involved in this kind of association often overcome the very concept of "religious pluralism" by constructing their own "spiritual way" beyond religious differences. We use the word "spiritual"[9] because it is a recurrent theme in the narration of couples' life stories and represents a major key to analyzing this reality. We found it at the center of the association's statutes and, also, as one of the keywords of the conferences that couples organized. This necessity to look for a common "spirituality" emerges as a common topic shared by the groups of couples taking part in the association's activities. Other couples progressively marginalize their former religious identities, removing any religious practice from the family context ("closeting" strategy), converting to the other partner's religion

---

[8]  So defined by the same association's leaders.

[9]  Although this article is not the place to summarise the long and controversial debate on the categories of 'spiritual and 'religious', it is important to briefly clarify the use of the terms and my position in the debate. However far the shift towards inner-life spirituality diagnosed by Houtman et al. (2012) and Campbell (2015) has advanced meanwhile, it is still a controversial issue that stimulates a rich debate around it. Some authors (Ammerman 2013; Zinnbauer et al. 1997) have contested the separation of the categories 'religious' and 'spiritual', arguing that the last should be understood as a moral rather than as an essential category. Houtman and Aupers (2007), analysing the data from the World Values Survey, interpreted the "neither-Christian-nor-secular outlooks" (Houtman et al. 2012, p. 29), suggested a New Age spirituality rather than "fuzzy fidelity" (Voas 2009, p. 167). Unlike Voas, they conceptualised these spiritualities as a third option beyond the common polarisation of traditional religions on the one hand and science, reason and secularism on the other. According to Houtman et al. (2012, p. 29), the term 'spiritual' is used as a "third corner of a triangle rather than a mixture of traditional theistic Christian religiosity and non-religiosity".

("conversion" strategy) or leaving to one partner the religious education of the children ("resigning" strategy). The couples involved in this association, instead, emphasize more on "spiritualization" focusing on the communalities between the faiths and downplaying the theological differences which can divide the couple.[10] In partners' life stories it emerges in the foreground of the attempt to construct a new dimension of faith, characterized by a personal relationship with the "sacred" which goes beyond dogmas. Amélie, herself the daughter of a mixed couple (Belgian father and Congolese mother), and Cédric, an immigrant from Congo, so explain their decision around children's religious education:

> I strongly believe in God. And I am a Christian. A Catholic. Now more than ever. [ . . . ] We first had a civil marriage in 1990 and then we married both in the mosque and in the church in 2010. That was also to have all our seven children there with us. That is to say how we deal with religions. Our children are not baptized but they can come to the Mass with me on Sunday, and they can pray with their father. That was an issue at the beginning. I was a bit worried not to baptize my children. But then I focused on the fact that this didn't mean I could not speak about my religion. And so, I did. And I think you do not perceive any conflict about this. But a same strong belief in God. And we are proud of this. [ . . . ] It is important to meet. It is important for us and for the new couples who need tips about how to deal with this thing. And not to worry about it. To hold on and go ahead with their relationship.

> Amélie, 58, Belgian.

> I never thought it would be a problem. It was really important for me to give a notion of my religion, Islam. [ . . . ] Christ is a central figure in Islam too. Most people don't know how much we have in common and how much our religions are in continuity. We live in a society almost atheist. It is way more important to focus on believing in God. That is something we both focused on. [ . . . ] We avoided choices that could have divided us. For instance, we chose for them not Muslim names but African names. To remind them of our common origins. [ . . . ] Our children and sometimes also my wife experienced fasting with me during Ramadan. We celebrate Christmas together. Got it. We share everything without problems. The spirituality is the important thing. To believe in God. That is one, the same. [ . . . ] With our association of mixed families, we try to testify this. And we share with other couples that it is possible to keep God in the family without any problem.

> Cédric, 59, Congolese.

Both Sarah and Cédric argue that their religious identities are an important part of themselves that they do not want to minimize or lose. The differences between their two religions are seen as something that concerns institutions rather than 'the same one God' in whom they both believe. Both partners narrate their mutual attempts at convergence, focusing on the need for a common desire to understand the other. That "strong" and "conflictual" difference often associated with the opposition between Islam and Christianity is here shifted to that between "believers" and "non-believers". The association's aim seems to be, as a consequence, to discuss a common effort to "keep God in the family".

The perennial message "God is love", often recurring during the association's meetings, emphasizes the mutual construction of a common representation of God beyond different world views. They overturn the hegemonic discourse on being "mixed" on the basis of religious denominations into a discourse on being similar on the basis of the common belief in God. To reach a compromise beyond the obstacles imposed by religious institutions seems to lead partners to create a new religious dimension within the family context where they create and share the importance of keeping a religious sensitivity, believing in God, beyond the barriers and what, for religious institutions, is perceived as

---

10　For a more detailed analysis of the different strategies adopted by partners in dealing with religious pluralism see Cerchiaro et al. (2015).

forbidden. A meaningful example is offered by these excerpts from Catherine and Amir, both active in the association for more than twenty years, from the time of their engagement.

> You have to find your way as a couple. We first married with civil marriage . . . then with an informal ceremony where there were, as normal guests, a priest, who read some Bible excerpts, and a Muslim scholar who read some excerpts from the Koran. [ . . . ]
>
> Catherine, 48, Belgian.

> We had three children. We decided not to baptize them or religiously mark them. We decided to pass on our common values and notions from both of us. At home. They did not follow religion at school, and they did not go to catechism. The association activities and the sharing of this path with other couples helped us to understand that it was possible to feed them in a spiritual way, without obliging them to choose between one or the other religion. We talk about how to live the faith in the couple, we go deeper into the Koran and the Bible, we dig out our religions instead of putting them aside. We pray together, and we talk about the education of children that is one of the main issues [ . . . ] Because it is easier to abandon religion or ask your wife to convert. But for us the path is to find another way to go to the core of our religions respecting each other . . .
>
> Amir, 49, Moroccan.

When this approach is adopted, both partners usually choose not to formally transmit one religion (in the case of Christianity, through baptism) and to manage at home the religious education of their children without delegating it to other educational institutions. Children thus represent a challenge that obliges partners to look for new mediations. Relatives and institutional actors (schools, the local church) influence and create obstacles to the already existing agreement between the partners and prompt them to seek a new balance. Amir explains how he and her partner try to balance the different interests and pressures on them from the outside world, keeping religious education as something privatized, personally managed by them without involving school or religious institutions.

The importance of the association emerges also for what concerns the socialization of their children. Through family meetings the couples try to create ties between their children and to give them the chance (and the space) to group and meet by themselves. Sarah, in the extract below, highlights the importance, for the children, of knowing their peers from similar families.

> The fact that we meet with other families like us, and that they know other children where their parents are Muslim and Christian, is important for them. It helps to create a sense of normality for them too. They feel they are not "strange". They understand that there are other peers with one parent Christian and the other Muslim. It is fundamental for them . . . I think this is one of the most important aspects of meeting with the association. For the children.
>
> Sarah, 41, Belgium.

The chance to share some time with other "mixed" families is described as an important vehicle for the "normalization" of their family, above all in relation to children. The association offers, thus, an environment to share experiences and share parenting advice but also the ideal location to bring children together and counter the sense of isolation and "strangeness".

*4.3. Social Activism*

The function of offering networks and support to the families is, indeed, often connected with the aim of being socially active. Ahmed summarizes below how the association of Christian-Muslim couples grew up within a wider association on Christian-Muslim relations. As one of the leaders of the group of Christian-Muslim couples he pointed out:

> With the Center for Christian-Muslim relations **[11] we organize a lot of social events about religious dialogue and for a deeper knowledge of Islam and Christianity. [ . . . ] The things they have in common . . . that are a lot . . . we focus on their spiritual aspects, their common aim to save man. To deepen the word of God . . . studying both the Koran and the Bible [ . . . ] Within this organization was born an association of Christian-Muslim couples. It is, thus, part of a wider reality. [ . . . ]

> Amir, 49, Moroccan.

Some of the couples involved are, indeed, active in promoting their life experience as an example for the wider society. The words of Cédric exemplify how partners' activism focuses on their vision of interreligious dialogue as the search for the common ground between Islam and Christianity. Cédric's involvement in local schools as a Muslim and as a partner in a mixed family became a whole, "a life mission" that aims to counter the dominant representation of Islam and Muslims.

> My aim is to use our life experience to teach people that we have to focus on what we have in common. I am also involved with schools. They call me to speak about my life experience to the students of high schools. I sometimes read passages of the Koran or of the Bible without saying what book I'm reading. [ . . . ] Then they become surprised when they understand the importance and great consideration that Jesus Christ and his mother have in the Koran. They just hear about terrorism and war, fanaticism [ . . . ] Our role in the society must be to show that we, Muslims and Christians, have a common spiritual background, common moral values . . . respect each other, peace, the importance of family . . . common principles that make us brothers and not enemies. This is our life mission.

> Cédric, 59, Congolese.

These activities testify to the social engagement aspect of the association that, through couples' meetings, tries to disseminate the experiences of these families to a wider group of people. Three main topics emerge from partners' narratives: (1) Migration and racism; (2) Christian-Muslim families and their quest for recognition; (3) Interconfessional events focused on a common spirituality.

Although it is not our aim, in this article, to analyze these events in depth, it is useful to briefly consider how they highlight the strict connection between the association's public and private spheres of activity.

The first topic includes all the initiatives and debates that are organized to sensitize a larger public to mixed families as examples of dialogue and coexistence between two cultures and two religions. Partners promote, for instance, thematic events based on food or journeys as occasion to enter in contact with other cultures. Events organized to deepen and discuss recent political fact concerning the issue of political asylum and the so-called "humanitarian emergence" are addressed during the association's meetings. Moreover, the theme of racism emerges starting from the direct experience of their children which are often considered as migrants or "second generations" because of their name and skin colour. As one partner said, "I've always raised my children saying, 'you don't live in a racist country'. But in the society, even in schools, they've always been associated with second generations even though they were born here and have a Belgian parent. And this is just for their black skin. So, we still have to work on this. And our families have to play an active role in the society. Fighting against the prejudice where the black skin is a synonym of non-European, non-Belgian".

The second topic includes those types of events where the association emerges as a social actor seeking more legitimacy and recognition within both Islamic and Christian religious communities. The decision of the couples to find a way of conciliating the different religious traditions and practices

---

[11]　In order to preserve the anonymity of our interviewees we omit the name of the organization.

around these issues often encounters various obstacle[12] posed by the respective religious institutions and authorities. The involvement of priests or imams in these events often produces resistance and denial, as testified by various narrations where partners recounted having experienced stigmatization and marginalization by religious institutions and their religious leaders. This is particularly true for couples where the Muslim partner is the woman. The Koranic prohibition on a Muslim woman to marry a non-Muslim man is opposed by the couples, who regard this prohibition as an obsolete piece of discrimination that should be overcome.

The third topic gravitates around the keyword "spirituality", which is declined as the overcoming of theological differences and the focus on the common aim and values of the "word of God". Couples introduce events like lectures and book presentations to personally deepen the perennial messages within Islam and Christianity. In this direction we can locate the events with some Sufi masters which help to convey a different and less known image of a more "spiritual" and "open" Islam. These meetings have the aim of proposing a different way of looking at religion and its role for individuals. These events offer insights that emphasize the common ground of the two religions (recurrent words in these events are thus: "the search for the truth", "inner peace", "purity" and "purification", "ecology" and "the role of Holy Mary" in Islam and Christianity).

## 5. Conclusions

This article seeks to contribute to the wider debate by offering a thicker understanding of the social consequences of mixedness (Rodriguez-Garcia 2015). Focusing on the case study of a "Christian-Muslim" families' association active in the Brussels capital region (Belgium), we highlight its activities, articulating our findings into three sections: "new couples", "parenting advisory" and "social activism". Through these data we can now return to our initial research questions: Why do couples turn to this association? At what stages of their lives? What is the social role that the association aims to play in the society?

The function of creating an environment to share and connect with other mixed families is the primary goal of the association. We highlight, in particular, how, for a newly formed couple, the association may represent an important place to face the first difficulties encountered by learning from older couples how they dealt with them. Some critical aspects common to all couples have been discussed: the opposition of the respective families, administrative issues, deciding between civil or religious marriage ceremonies and all the initial management of religious and cultural difference. We then focused on the centrality of parenting as a key issue that pushes the couples to find a way to construct what I define as "the spiritual way", a symbolic path that the couples narrate as a new way of living and transmitting religious values beyond an institutionalized definition of religion. In doing this, the article problematizes the assumptions that interpret these marriages as resulting only from or in secularization or conversion processes. The life stories of these couples show us that, in the effort to preserve a religious dimension, partners can reach a new shared equilibrium which, de facto, overcomes religious pluralism. Our data shows that there is not necessarily a "loosening" (Voas 2003) or a "dilution" (Song and Gutierrez 2015b) of partners' respective religious identities but, rather, a reshaping or reconfiguration of them. Children represent a challenge that obliges partners to look for new mediations. The daily family life becomes a space for mixing different religious practices and an incentive to personally deepen the interpretation of God's words in "The Book". In this "spiritual way", as previously suggested in Cerchiaro et al. (2015), according to religious individualization theory and despite secularization theory, there is no necessary correlation with a loss of religiousness for the

---

12　Just to mention those to which we are referring: the declaration of Christian faith expected in the Christian marriage service; the difficulty in obtaining a dispensation for disparity of cult (in my wider research reported above all by couples who live in Italy); the presence only of two male Muslim witnesses being required in Islamic marriage; the expectations, for both Islam and Christianity, that the parents will raise their children within their respective religions and, as mentioned above, the prohibition for Muslim women to marry non-Muslim men.

individual. On the contrary, more subjective and privatized forms of religion, like those which our interviewed couples are trying to create, are replacing institutionalized ones (Roof 1999; Houtman and Mascini 2002; Houtman and Aupers 2007; Pollack and Pickel 2007; Houtman 2016).

Finally, the article tries to deepen our understanding of what kind of social activism this association expresses. Partners involved seek recognition and legitimacy from the same religious institutions by which they feel rejected or marginalized. The focus on the common "spiritual" ground of the religions, together with the focus on "truth" and "purity" (Roeland et al. 2010) demonstrates the construction of a kind of "new vocabulary" used to speak about religion. Moreover, the findings reveal that this association works to help partners to maintain their religious roots, reinterpreting them dynamically, in accordance with their family status. The meetings and events focused on the common points between the Bible and the Koran testify to how partners active in the association work as agents of interreligious dialogue. Their focus on religious identities perceived not as fixed entities but, rather, as an inner process in constant evolution, should be interpreted not as a "believing without belonging" (Davie 1990) but rather as a quest for change and renovation in religious institutions. Partners, thus, construct new "plausibility systems" intended as a "purer" interpretation of religious teachings which overcome the religious dogma and rules that claim to determine their sentimental and parental choices.

**Funding:** This research was funded by the Marie Skłodowska-Curie actions, within the European Commission Research and Innovation programme, Horizon 2020. Funding scheme: MSCA-IF-EF-ST. Grant number 747592. Proposal acronym: ReMix. Proposal title: Christian-Muslim families dealing with religious pluralism in everyday family life Religious reconstruction in religiously mixed marriages

**Conflicts of Interest:** The author declares no conflict of interest.

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
