# Peer review of "Fighting for What? Couples’ Communication, Parenting and Social Activism: The Case Study of a “Christian-Muslim” Families’ Association in Brussels (Belgium)"

_religions, doi:10.3390/rel10040270_

Round 1

Reviewer 1 Report

I found this to be an original approach to an age-old question.  I see a limitation of the research in that all the men were originally Muslim and the women Christian.  A nice comparison group would be Muslim women and Christian men (if there are some to study). 

Author Response

Dear Reviewer,

I would like to thank you for the time dedicated to my article.

The article presented the first results from my current Marie Curie project. It represents, therefore, an ongoing project on which I am still working. Since the date I submitted the article I have interviewed another couple part of this association who is comprised of a Muslim woman and a non-Muslim man. The comparison was not yet possible since 12 couples are characterized by Muslim men and just one by a Muslim woman. It was anyway beyond the scope of the article to address this topic. As I have now pointed out it in the methodology section, the issue of Muslim-women with non-Muslim men has not been discussed yet and deserves a separate analysis because of its social, theological and legal implications. It was stated that this represents a limitation of the article.

By focusing on the case study of an association of “Christian-Muslim” families in Belgium, my aim was here restricted to the following research questions: Why do couples turn to this association? At what stages of their lives? What is the social role that the association aims to play in society? I think the article offers a first study on one of this association and it is, of course, limited to answer to this research questions while other research questions, as the one highlighted by the reviewer needs to be further elaborated.

Once again, thank you for raising this issue and giving me the opportunity to improve this aspect of my article.

Best regards,

The Author

Reviewer 2 Report

This is a focused and interesting piece of research, conceptually sophisticated, elegantly written  and engaging with a wide range of relevant secondary literature. It's conclusions also nuances and challenges the theoretical debates around such mixed marriages - whether supposed  'dilution' or 'loosening' or religious and ethnic ties. A model article.

Author Response

Dear Reviewer,

I would like to thank you for the time dedicated to my article.

I have revised and improved some sentences and clarified some methodological issues about the participants to my research.

Once again, thank you for giving me the opportunity to improve my article.

Best regards,

The Author